# Prompt Tuning for Multi-Label Text Classification: How to Link Exercises to Knowledge Concepts?

Liting Wei [1,2], Yun Li [1,*], Yi Zhu [1], Bin Li [1] and Lejun Zhang [1]

1   School of Information Engineering, Yangzhou University, Yangzhou 225012, China
2   Jiangsu Provincial Key Constructive Laboratory for Big Data of Psychology and Cognitive Science,
    Yancheng Teachers University, Yancheng 224002, China
*   Correspondence: liyun@yzu.edu.cn; Tel.: +86-13852785198

**Abstract:** Exercises refer to the evaluation metric of whether students have mastered specific knowledge concepts. Linking exercises to knowledge concepts is an important foundation in multiple disciplines such as intelligent education, which represents the multi-label text classification problem in essence. However, most existing methods do not take the automatic linking of exercises to knowledge concepts into consideration. In addition, most of the widely used approaches in multi-label text classification require large amounts of training data for model optimization, which is usually time-consuming and labour-intensive in real-world scenarios. To address these problems, we propose a prompt tuning method for multi-label text classification, which can address the problem of the number of labelled exercises being small due to the lack of specialized expertise. Specifically, the relevance scores of exercise content and knowledge concepts are learned by a prompt tuning model with a unified template, and then the multiple associated knowledge concepts are selected with a threshold. An Exercises–Concepts dataset of the Data Structure course is constructed to verify the effectiveness of our proposed method. Extensive experimental results confirm our proposed method outperforms other state-of-the-art baselines by up to 35.53% and 41.78% in Micro and Macro F1, respectively.

**Keywords:** linking exercises to concepts; multi-label text classification; prompt tuning; few-shot

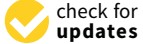



## 1. Introduction

In recent decades, personalized learning has become a mainstream solution to enhance students' learning interest, and experience in intelligent education systems [1–3]. One of the fundamental and key tasks in personalized learning is knowledge tracing [4,5], which aims to evaluate the students' learning state of knowledge concepts.

Exercises have played an important role in the knowledge tracing tasks, which is one of the evaluation metrics of whether students have mastered specific knowledge concepts [6,7]. Students in intelligent education systems choose the right exercises according to their own needs and acquire specific knowledge concepts during exercise. In turn, we can track changes in students' acquisition of knowledge concepts during their exercising process. From this perspective, knowledge tracing should consist of a students–exercises–knowledge concepts hierarchy [8]. However, most existing methods of knowledge tracing approaches [9–11] are partially modeled among the hierarchy (i.e., students–exercises or students–concepts). This is because, in some intelligent systems, there is a lack of connection between exercises and knowledge concepts. To this end, we take the automatic linking of exercises to knowledge concepts into consideration for knowledge tracing tasks.

In essence, linking exercises to knowledge concepts is a multi-label text classification (MLTC) problem. As shown in Figure 1, the relationship between exercises and knowledge concepts is one-to-one or one-to-many, which aims to assign one or more concepts to each input exercise in the dataset. Moreover, Figure 1 shows that the semantics between exercises and knowledge concepts are highly correlated.

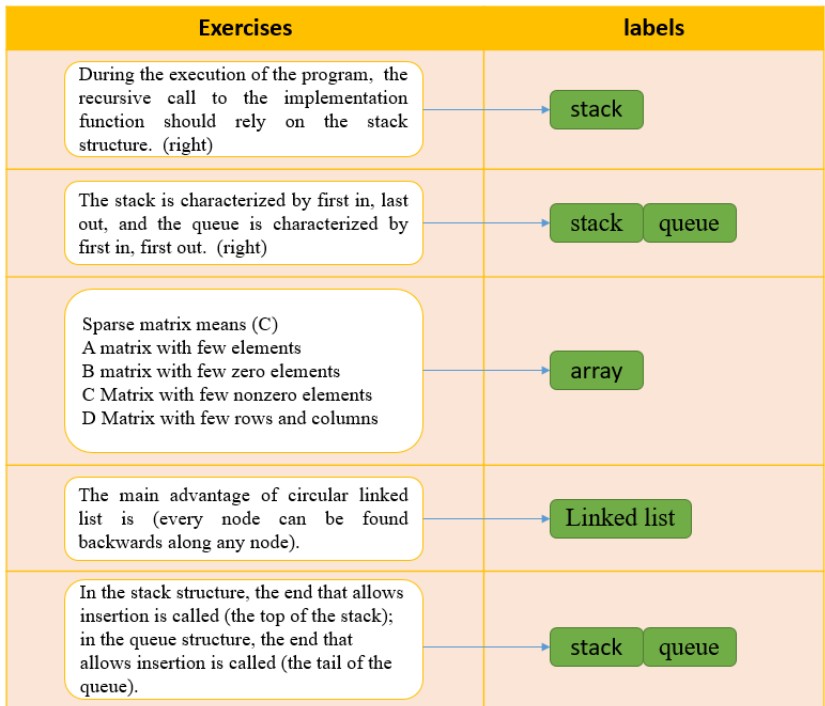

**Figure 1.** Examples of exercises linking to knowledge concepts from dataset.

Recently, deep-learning-based methods have achieved fairly good performance in MLTC for the superiority of feature representation learning. For example, Liu et al. [12] utilized the strengths of the existing convolutional neural network and took multi-label co-occurrence patterns into account in the optimization objective to produce good results in MLTC. Pal et al. [13] proposed a graph attention network-based model to capture the attentive dependency structure among the labels. Chang et al. [14] fine-tuned the BERT language model [15] to capture the contextual relations between input text and the induced label clusters. However, these deep-learning-based methods in MLTC tasks require large amounts of training data for model optimization, which is usually time-consuming and labour-intensive in real-world scenarios. Unfortunately, linking exercises to knowledge concepts usually lacks training data because some knowledge concepts corresponding to a few exercises or new courses may contain a paucity of labelled data.

To address these problems, we propose a Prompt Tuning method for Multi-Label Text classification (PTMLTC for short). First, the prompt tuning model with a unified template predicts the relevance scores of exercises and knowledge concepts. Then, the multiple associated knowledge concepts are picked with a threshold. In order to verify the effectiveness of our proposed method, an Exercises–Concepts dataset of the Data Structure course is constructed. Extensive experimental results confirm our method outperforms other state-of-the-art methods by up to 32.53% and 41.78% in Micro and Macro F1, respectively.

The contribution of our paper can be summarized as follows:

(1) To the best of our knowledge, this is the first attempt to automatically link exercises to knowledge concepts. We built an Exercises–Concepts dataset of the Data Structure course and reconstructed the few-shot dataset.

(2) We propose a prompt tuning method for multi-label text classification to link exercises to knowledge concepts. Large amounts of labelled or unlabeled training data are not required.

(3) Extensive experimental results confirm that our proposed method outperforms other state-of-the-art deep-learning-based methods.

## 2. Related Work

In this section, firstly, we introduce the deep-learning-based multi-label text classification methods. Then, the prompt tuning learning methods used in our models will be presented.

### 2.1. Multi-Label Text Classification

The goal of MLTC is to associate one or more relevant labels for each input text instance. The traditional MLTC methods include one-vs-all methods [16,17], tree-based methods [18,19] and embedding-based methods [20,21]. For example, Babbar et al. [16] proposed a distributed learning mechanism for MLTC, which can use doubly parallel training to reduce the expensive computational cost of one-vs-all methods. Prabhu et al. [22] presented a method called FastXML by optimizing an nDCG-based ranking loss function to further reduce expensive computational costs. Tagami [21] proposed a graph embedding method, which learns partition data points by the k-nearest neighbour graph (KNNG) and uses an approximate k-nearest neighbour to predict results by exploring KNNG in the embedding space.

In recent years, due to the powerful ability of feature representations learning [23,24], deep models have gained much attention and achieved superior performances over traditional methods. The focus of existing deep-learning-based methods on MLTC is learning-enhanced text representation for improving performance. For example, Liu et al. [12] utilized the strengths of the existing convolutional neural network (CNN) and dynamic pooling to model the text representation for MLTC. Xiao et al. [25] employed an attention mechanism to explore highlight important context representation in MLTC tasks. Ma et al. [26] utilized the bidirectional Gated Recurrent Unit network and hybrid embedding for learning the representation of the text level-by-level. Chang et al. [14] proposed to fine-tune the BERT language model [15] in order to capture the contextual relations between input text for MLTC.

In addition, recently, the dependencies or correlations among labels have demonstrated the ability to improve performance in most MLTC tasks. Along this line, many deep-learning-based methods have been proposed to model label dependencies. For example, Chen et al. [27] explored labels' correlations through Recurrent Neural Networks, which were used to predict labels one-by-one sequentially. Pal et al. [13] proposed a graph-attention network-based model to capture the attentive dependency structure among the labels. Yang et al. [28] treated MLTC tasks as a sequence generation problem and proposed a decoder structure to capture the dependencies between labels that selected the most informative words automatically while predicting different labels. Xun et al. [29] learned label correlation by introducing an extra CorNet module that is applied to a deep model at the prediction layer to enhance raw label predictions with correlation knowledge.

However, most existing deep-based MLTC methods require a large amount of labelled or unlabeled training data for model optimization, which is often time-consuming and labour-intensive. Therefore, designing methods that can achieve promising results in the few-shot scenario remain a huge challenge in real-world MLTC tasks.

### 2.2. Prompt Tuning

Prompt-based learning [30–32] is regarded as a new paradigm in natural language processing and has drawn great attention from multiple disciplines, which promotes the downstream tasks by using the pre-training knowledge as much as possible. Starting from the GPT-3 [33], Prompt tuning has demonstrated unique strengths in a variety of tasks, which contain text classification [32,34], relation extraction [35], event extraction [36] and so on. Prompt-based learning directly models the probability of text on top of language models. It is different from traditional supervised learning, which trains a model to predict the output y as $P(x \mid y)$ with the input $x$. Specifically, in the prediction task, firstly, a template is added to the original input $x$ to form a new textual string prompt $x'$ with [MASK]. Then, the reconstructed $\hat{x}$ is learned with the language model to probabilistically

fill the unfilled information. For example, Cui et al. [37] employed closed prompts filled by a candidate named entity span as the target sequence in named entity recognition tasks. Li et al. [38] proposed Prefix-tuning that uses continuous templates to improve performance than discrete prompts. There has already been some recent effort in devoting external knowledge to prompt design. For example, Hu et al. [34] proposed a knowledgeable prompt-tuning by expanding the label word space of the verbalizer with external knowledge bases. Chen et al. [35] proposed a knowledge-aware prompt-tuning approach, which introduced relation labels knowledge into prompt construction. In addition, many works [34,39] have demonstrated that prompt-based learning greatly improves model performance in few-shot scenarios. Hambardzumyan et al. [40] proposed an automatic prompt generation method to transfer knowledge from large Pre-trained Language Models, which achieved excellent performance in a few-shot setting. Gu et al. [41] proposed to add soft prompts into the pre-training stage and pre-train soft prompts in the form of unified classification tasks, which can reach or even outperform in few-shot settings. However, in the knowledge tracing tasks, we are not aware of existing prompt-learning-based approaches that automatically link exercises to knowledge concepts. To this end, we propose a prompt tuning method for multi-label text classification to link exercises to knowledge concepts.

## 3. Prompt Tuning Method for Multi-Label Text Classification

In this section, the details of our proposed PTMLTC are given, and the general framework is shown in Figure 2.

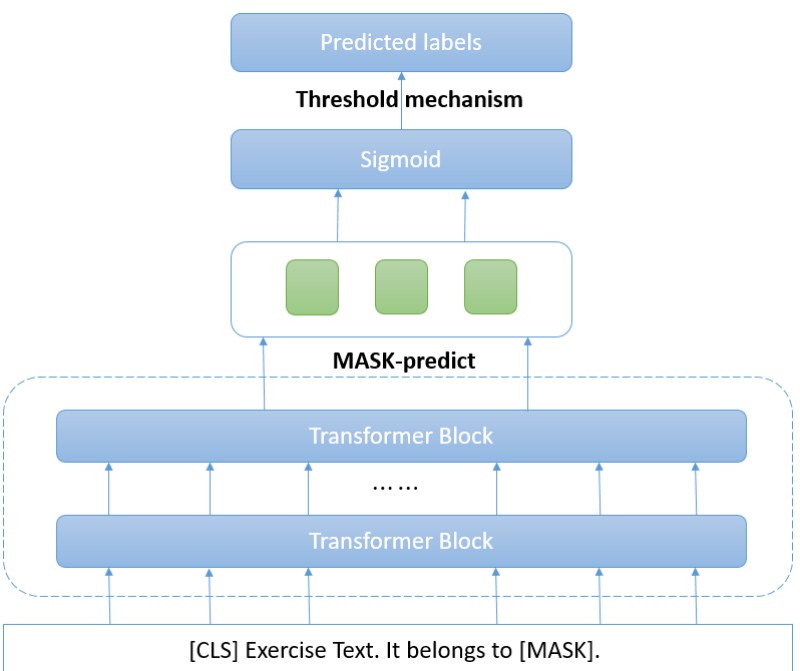

**Figure 2.** The general framework of our PTMLTC. Exercise Text sequence is connected with united template as the input of prefix Language Model. It will then predict the probability of filling the token [MASK] with each word of knowledge concepts. $Sigmoid()$ function is used to obtain the probability of exercise texts linking to knowledge concept labels. Finally, a threshold mechanism is adopted to predict all the possible knowledge concept labels.

### 3.1. Problem Formalization

In this paper, we aim to use few exercises with labeled concepts to predict one or more related concepts for each input exercise text. Given $C = \{c_1, c_2, \cdots c_N\}$ is the label space with $N$ concepts, the goal is to learn a function $h(\cdot) : E \rightarrow 2^C$ from the support set $S = \{(E_i, C_i)\}_i^{N_S}$, where $E$ denotes the exercise-instance space, $S$ usually contains $K$ exercise-instances ($K$-shot) of $N$ concept-labels ($N$-way), $N_S$ is the size of the support set.

For each learning instance $(E_i, C_i)$, $E_i \subseteq E$ is $l$-dimensional input and $C_i \subseteq C$ is the related concepts set. For an unseen instance $e$ in the query set, the classifier predicts a set of concepts $P = h(e) \subseteq C$.

### 3.2. Prompt Tuning Method for Multi-Label Text Classification

As is shown in Figure 2, our methods adopt a threshold-based strategy [42,43] to achieve multi-label text classification. Firstly, the relevance scores of exercise content and knowledge concepts are transformed into a masked language model by prompt tuning methods. Specifically, a prompt template is defined as $V_{prompt} = "It\ belongs\ to\ [MASK]"$. and combine the exercise text $x = \{x_0, x_1, x_2, \cdots, x_n\}$ to form the final input for prompt tuning input $e_{prompt}$, which can be shown as Equation (1):

$$e_{prompt} = [CLS]x, It\ belongs\ to\ [MASK]. \tag{1}$$

Suppose that $M$ is a large corpus of Pre-trained Language Models (PLMs in short), the probability of filling the token [MASK] for each word of concept $c$ in the knowledge concepts set $C$ can be denoted as $P_M([MASK] = c \mid e_{prompt})$. Here, we need a map function $Sigmoid()$ to predict the probability of each concept independently. The relevance scores can be represented as (2):

$$P(c \mid e_{Prompt}) = Sigmoid(P_M([MASK] = c \mid e_{prompt})) \tag{2}$$

Finally, we add an additional threshold mechanism to determine knowledge concepts corresponding to exercises, which can be formulated as (3):

$$P(e) = \{c \mid P(c \mid e_{Prompt}) > t, c \in C\} \tag{3}$$

where $t$ is the threshold.

To better introduce our method, we take an example shown in Figure 3. The exercise text "The stack is characterized by first in, last out, and the queue is characterized by first in, first out. (right)" is wrapped with template as the input. PLM is adopted to predict the predict the probability of filling the token [MASK] with knowledge concepts word set array, stack, queue, linked list. Then, $Sigmoid()$ function is used to obtain the probability of exercise text linking to labels $\{array, stack, queue, linkedlist\}$. Due to the probability of exercise text linking to stack, queue greater than threshold, exercise text is regarded as linking to $\{stack, queue\}$.

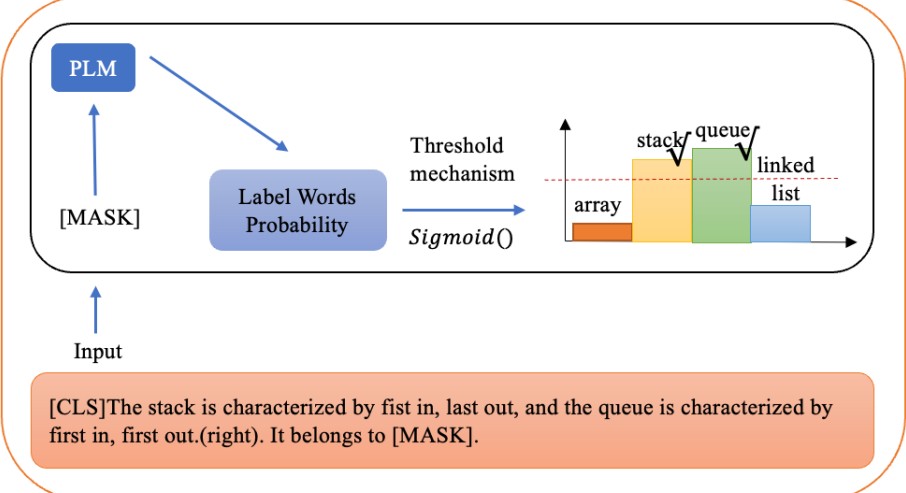

**Figure 3.** An example of our proposed method.

It has been proven that binary cross-entropy loss (BCE) over sigmoid activation is more suited for multi-label problems and outperforms cross-entropy loss [12]. Therefore, in our paper, the BCE loss function is chosen to learn parameters in the tasks, which can be formulated as (4):

$$\min_{\Theta} -\frac{1}{K} \sum_{i=1}^{K} \sum_{j=1}^{N} [y_{ij} \log(\sigma(\hat{p}_{ij})) + (1 - y_{ij}) \log(1 - \sigma(\hat{p}_{ij}))] \tag{4}$$

where $\hat{p}_{ij}$ represents the predicted value of exercise $i$ belongs to concept $j$. $y_{ij}$ represents the value of exercise $i$ belongs to concept $j$, and $\sigma$ is the sigmoid function $\sigma(x) = \frac{1}{1+e^{-x}}$.

## 4. Experiment

In this section, we conduct extensive experiments on the constructed Exercises–Concepts dataset of the Data Structure course to verify the effectiveness of our proposed method for linking exercises to knowledge concepts. In the following, firstly, the Exercises–Concepts dataset of the Data Structure course and the few-shot dataset construction are introduced in detail. Then the compared methods and evaluation metrics of our experiments are shown. Finally, we analyze the experimental results and the influence of the main parameters.

### 4.1. Datasets

Exercises–Concepts dataset of Data Structure course: To study the problem of linking exercises to knowledge concepts, we construct the Exercises–Concepts dataset of the Data Structure course. Refer to MOOCCube_DS [44] data repository and Several national planning textbooks, we extract 65 classic knowledge concepts. Subsequently, 2027 exercises used in these textbooks are marked with the corresponding knowledge concepts. Details are shown in Table 1.

Few-shot dataset construction: To simulate the few-shot situation, we reconstruct the dataset in to the form of few-shot learning, where each example is the combination of a query instance $(e^q, c^q)$ and the corresponding $K - shot$ support set $S$. Unlike the single-label classification problem, instances of multi-label classification may be associated with multiple labels. Therefore, there is no guarantee that each label appears exactly $K$ times during sampling. To address the problem, we approximately construct $K - shot$ support set $S$ with the Minimum-including Algorithm [43]. It constructs a support set generally complying with the following two conditions: (1) All labels in the original dataset appear at least $K$ times in support set $S$. (2) At least one label will appear less than K times in $S$ if any $(e^q, c^q)$ pair is removed from it. For the original dataset, we sampled $N_S$ different support sets. For each support set, we take the remaining data as the query set. Each support-query-set pair constitutes one few-shot episode.

On the test stage, we constructed 10 different few-shot episodes for each selected K-shot. Among them, support set is used to fine tuning model, and query set is used to test the effectiveness of methods.

**Table 1.** The number of each exercise linking to knowledge concepts. *C* represents the label of the knowledge concept, *N* denotes the number of exercise.

| C | N | C | N | C | N | C | N |
|---|---|---|---|---|---|---|---|
| Array | 36 | Sequential List | 46 | Bubble Sort | 41 | Binary Search | 48 |
| Logical Structure | 48 | Linked Storage Structure | 48 | Time Complexity | 91 | Generalized table | 35 |
| Heapsort | 46 | Physical Structure | 35 | Linked List | 10 | Matrix | 35 |
| Adjacency Matrix | 35 | Linear List | 48 | BT-Preorder Traversal | 37 | Tree-Degree | 40 |
| Algorithm | 56 | String | 35 | BT-Postorder Traversal | 35 | Tree-Depth | 45 |
| Queue | 35 | Tree | 24 | Minimum Spanning Tree | 53 | Graph | 18 |
| Recursion | 39 | Binary Tree (BT) | 23 | Topological Sort | 45 | Circular Queue | 35 |
| Complete Binary Tree | 42 | Binary Sort Tree | 35 | Depth First Search | 5 | Binary Tree-Threaded BinaryTree | 35 |
| Balanced BinaryTree | 50 | Huffman Tree | 35 | Breadth First Search | 48 | Shell's Sort | 60 |
| Search | 136 | Data Structure | 35 | Connected Graph | 47 | Binary Tree-Inorder Traversal | 13 |
| Sequential Search | 53 | Sequential Storage Structure | 35 | Quick Sort | 68 | Merge Sort | 35 |
| Critical Path | 60 | Stack | 35 | Full Binary Tree | 48 | Space Complexity | 35 |
| Selection Sort | 35 | Strongly Connected Graph | 57 | Graph-Degree | 35 | Selection Sort | 35 |
| HashSearch | 35 | Muitl-way Search Tree | 35 | Adjacency List | 37 | Sort | 28 |
| Shortest Path | 35 | Binary Tree-Order Traversal | 56 | Doubly Linked List | 35 | Straight Insertion Sort | 44 |
| Cycle Chain | 14 | Undirected Graph | 27 | Oriented graph | 36 | Data | 12 |
| Double Circle List | 8 | | | | | | |



*4.2. Baselines and Evaluation*

4.2.1. Baselines

Traditional deep-learning-based multi-label text classification methods, such as XML-CNN [12], MAGNET [13], require massive amounts of training data for model optimization, which inevitably leads to performance degradation in the few-shot scenario. However, the PLMs tuning multi-label text classification methods can provide a certain advantage in the few-shot problem. Therefore, three PLMs tuning methods are conducted as compared methods, the details are described as follows:

TextCNN [45]: The method uses a simple CNN with one layer of convolution on top of word vectors for Sentence Classification. In our experiments, PLMs are used to learn the representation of words, in addition, a multi-label classification layer is added to predict labels. Notably, the method is fine-tuned on the support set to select the optimal model and validated on the query set.

TagBert [46]: This is a model based on a large pre-trained model and a multi-label classification layer. Following the parameter setting of a threshold-based multi-label method, a fixed threshold tuned on the support set is used in the experiments.

BertFGM (https://github.com/percent4/keras_bert_multi_label_cls (accessed on 2 April 2021)): Based on the TagBert method, adversarial training [47] is introduced to increase the robustness and generalization of the model.

The experimental setup of all the above methods is the same as that in TextCNN.

4.2.2. Evaluation

In our paper, the *MacroF*1 and *MicroF*1 are introduced to evaluate the effectiveness of our proposed method. *MacroF*1 calculates the average of the *F*1 scores obtained for each category, which can be formulated as (5):

$$
\begin{aligned}
P_t &= \frac{TP_t}{TP_t + FP_t} \\
R_t &= \frac{TP_t}{TP_t + FN_t} \\
Macro\ F1 &= \frac{1}{\mid C \mid} \sum_{t \in C} \frac{2P_t R_t}{P_t + R_t}
\end{aligned}
\tag{5}
$$

where $P_t$ represents the precision of each category, $R_t$ represents the recall of each category. $TP_t$, $FP_t$ and $FN_t$ are the true-positive, false-positive and false-negative example of the $t$-th label in the label set $C$, respectively. *MicroF*1 calculates the overall of the *F*1 scores, which can be formulated as (6):

$$
\begin{aligned}
P &= \frac{\sum_{t \in C} TP_t}{\sum_{t \in Y} TP_t + FP_t} \\
R &= \frac{\sum_{t \in C} TP_t}{\sum_{t \in C} TP_t + FN_t} \\
Micro\ F_1 &= \frac{2PR}{P + R}
\end{aligned}
\tag{6}
$$

where $P$ represents the overall precision, $R$ represents the overall recall.

*4.3. Experimental Results*

4.3.1. Experiment Settings

We evaluate the performance of our proposed method on the few-shot Exercises–Concepts dataset. Because some concepts in the dataset have only 5 exercises, we select the value $K$ in K-shot as 1 and 5, respectively. There are some hyper-parameters that need to be initialized in the above methods. Firstly, we introduce uniform settings in all methods. The maximum length sequence is set as 512. These models are optimized by Adam with batch size 4 and learning rate $1 \times 10^{-5}$. Then, the size of thresholds has an impact on final performance. The thresholds are set as 0.10, 0.65, 0.82, 0.24 on 1-shot setting in TextCNN,

TagBert, BertFGM and PTMLTC, respectively. On 5-shot setting, the thresholds are 0.08, 0.70, 0.85 and 0.20. The reported results are the mean and variance of the experimental results on 10 randomly generated few-shot datasets.

### 4.3.2. Performance Comparison

Results of 1-shot setting:The results of the 1-shot exercise linking to knowledge concepts are shown in Table 2. From the experimental results, we can have the following observations. Firstly, we can observe that the results of *MicroF*1 and *MacroF*1 in the PTMLTC method are 54.74% and 46.11%, respectively, which are far better than the other three baselines. In the case of much training data, the performance of BertFGM is better than the TagBert. However, added adversarial training in the few-shot problem obtains interference information, which makes the classifier more indistinguishable. BertFGM achieves worse results than the TagBert. Results of 5-shot setting: The results of the 5-shot exercise linking to knowledge concepts are shown in Table 3. The results are basically consistent with the trend of the 1-shot setting. Compared with the 1-shot setting, the results of all methods have been improved in the 5-shot setting. These results demonstrated that the increasing of training data improves classification performance. In addition, PTMLTC has a smaller margin of advantage in 5-shot setting compared with 1-shot setting. It is proved that the fewer the data, the more obvious the advantages of PTMLTC.

**Table 2.** Results of 1-shot on our dataset. Metrics marked in bold contain the highest metrics for the dataset.

| Method | 1-Shot | |
| --- | --- | --- |
| | Micro F1 | Macro F1 |
| TextCNN | $6.60 \pm 1.23$ | $5.70 \pm 0.89$ |
| TagBert | $9.83 \pm 0.77$ | $6.05 \pm 2.16$ |
| BertFGM | $6.65 \pm 2.47$ | $2.11 \pm 2.13$ |
| PTMLTC | $53.86 \pm 3.16$ | $\mathbf{49.04 \pm 3.42}$ |

**Table 3.** Results of 5-shot on our dataset. Metrics marked in bold contain the highest metrics for the dataset.

| Method | 5-Shot | |
| --- | --- | --- |
| | Micro_F1 | Macro_F1 |
| TextCNN | $29.49 \pm 0.62$ | $29.84 \pm 2.67$ |
| TagBert | $47.06 \pm 0.18$ | $41.50 \pm 6.90$ |
| BertFGM | $34.72 \pm 0.99$ | $26.66 \pm 3.12$ |
| PTMLTC | $\mathbf{62.37 \pm 0.43}$ | $\mathbf{58.84 \pm 0.84}$ |

### 4.3.3. Ablation Study

We compare the effects with different PLMs. In our proposed methods, Bert [15] and Roberta [48] models are adopted with bert-base-chinese (https://huggingface.co/bert-base-chinese (accessed on 5 February 2022)) and chinese-roberta-wwm-ext (https://huggingface.co/hfl/chinese-roberta-wwm-ext (accessed on 6 February 2022)). Table 4 summarizes The results are summarized in Table 4, which shows the Roberta-based pre-training model achieves better results than Bert.

The success of prompt tuning mainly owes to the template design and label words. Different templates are designed in our method to discuss their effect. The details are shown Table 5. The template was selected as "It belongs to [MASK]", which obtains the better result.

**Table 4.** Results of different PLMs on our dataset. Metrics marked in bold contain the highest metrics for the dataset.

| Method | Micro F1 | | Macro F1 | |
| --- | --- | --- | --- | --- |
| | **1-Shot** | **5-Shot** | **1-Shot** | **5-Shot** |
| PTMLTC_Bert | 50.74 ± 1.53 | 58.56 ± 1.50 | 46.11 ± 2.68 | 54.28 ± 1.47 |
| PTMLTC_Roberta | **53.86 ± 3.16** | **62.37 ± 0.43** | **49.04 ± 3.42** | **58.84 ± 0.84** |

**Table 5.** Results of the different design of the templates. Metrics marked in bold contain the highest metrics for the dataset.

| Templates | 1-Shot | | 5-Shot | |
| --- | --- | --- | --- | --- |
| | **Micro F1** | **Micro F1** | **Micro F1** | **Micro F1** |
| It belongs to [MASK]. | **53.86 ± 2.74** | **49.04 ± 3.12** | **62.37 ± 1.05** | **58.84 ± 0.89** |
| The concept is [MASK]. | 50.98 ± 2.92 | 51.35 ± 2.15 | 58.44 ± 0.77 | 54.28 ± 1.15 |
| The concept belongs to [MASK]. | 52.76 ± 3.13 | 46.83 ± 2.47 | 60.99 ± 0.37 | 53.85 ± 0.62 |

### 4.3.4. Parameter Sensitivity

Regarding our proposed method, in this section we have studied the influence of the parameter, which is the threshold $t$ in Equation (3). The experimental mode of control variables is adopted, when one variable is changed, the other variables remain unchanged. We randomly selected a dataset from the 1-shot and 5-shot few-shot datasets for verification. After some preliminary tests, we found that the value of $t$ will have a relatively large impact on the effect, it can be ensured that the effect will not excessively fluctuate within a certain range. The value set of t is [0.14, 0.16, 0.18, 0.22, 0.24, 0.26]. It can be observed from Figure 4 that $t = 0.24$ on the 1-shot setting and $t = 0.2$ on the 5-shot setting lead to the best results.

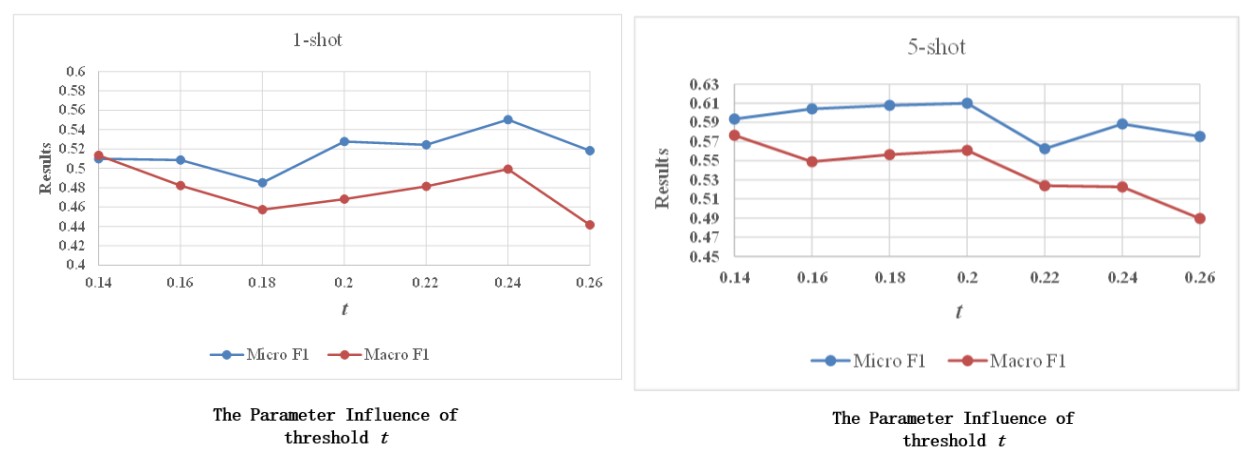

**Figure 4.** Effects of threshold $t$ on two datasets.

## 5. Conclusions and Future Work

In this paper, a prompt tuning multi-label text classification method is proposed to realize the link between exercises and knowledge concepts. The main idea is that the relevance scores of exercise content and knowledge concepts are learned by a prompt tuning model with a unified template, and then the multiple associated knowledge concepts are selected with a threshold. On the constructed dataset, we compare the proposed method with other baseline methods. The results show that PTMLTC achieves better performance than other state-of-the-art methods in the evaluation metrics, and with fewer training data, the advantage is more conspicuous. The knowledge concepts in the course bear a natural

graph relationship, and our work ignores the relationship between them. Future work will try to introduce the structural relationship between knowledge concepts into the model for achieving better results.

**Author Contributions:** Methodology, Y.L.; Resources, B.L.; Validation, L.Z.; Writing—original draft, L.W.; Writing—review & editing, Y.Z. All authors have read and agreed to the published version of the manuscript.

**Funding:** This research received no external funding.

**Institutional Review Board Statement:** Not applicable.

**Informed Consent Statement:** Not applicable.

**Data Availability Statement:** Not applicable.

**Conflicts of Interest:** The authors declare no conflict of interest. The funders had no role in the design of the study; in the collection, analyses, or interpretation of data; in the writing of the manuscript, or in the decision to publish the results.

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
