# Peer review of "Prompt Tuning for Multi-Label Text Classification: How to Link Exercises to Knowledge Concepts?"

_applsci, doi:10.3390/app122010363_

Round 1

Reviewer 1 Report

The paper studies the multi-label text classification task to link exercise to knowledge concepts with prompt tuning techniques. The motivation of the problem is good, and the method is also reasonable. The experiments also show the designed algorithm provides significant improvements.

However, there are some issues to be clarified or improved.

  1. The primary technique used in this paper seems to be prompt. But, the authors didn’t describe the design of the prompt, whether there are different choices for prompts, and how to choose which prompt to use. This makes the paper look lacking in technical depth.
  2. For experiments, maybe the ablation tests can also be used to test different choices for prompts.
  3. For experiments, can the test also be performed on more datasets, not only the data structure dataset?
  4. Some typos need to be fixed. Function sigmod above equation 2? Equations 5 and 6 should be swapped. Typos in Table 1.

Reviewer 2 Report

The paper proposes a method for multi-label text classification, in which it addresses the problem of the relevance scores of content and the knowledge concepts are learned by a model with a unified template. 

My observations/recommendations regarding the work are the following:

- The predicted labels phase (which appears in Fig. 2) is not described in detail, so I would recommend a more extensive description of it.

- The unified template as input, from in formula 1 does not has described the components that make it up

- The proposed method should be described in more detail, perhaps by exemplifying a smaller data set

- In Formulas 5 and 6, their component elements are not described (what each element represents?) and how they are applied in the case of a small data set

- The data set on which the method was tested, the size, and other details are not specified. It would be important to clarify.

Round 2

Reviewer 1 Report

The authors have addressed all my comments. no further comments. 

Author Response

Dear reviewer:

Thanks for your comments.

Yours Sincerely

Reviewer 2 Report

The work was improved considering the observations made. However, I propose some recommendations to be considered:

- Figure 3 is not loaded correctly and clearly visible in the final document

- In figure 4, the measurement units for the Ox and Oy axes are not specified

- The conclusions part must highlight the advantages of the proposed method compared to other similar methods in the field
